# Emergency Medical Services (EMS) Transportation of Trauma Patients by Geographic Locations and In-Hospital Outcomes: Experience from Qatar

**DOI:** 10.3390/ijerph18084016

**Published:** 2021-04-12

**Authors:** Hassan Al-Thani, Ahammed Mekkodathil, Attila J. Hertelendy, Ian Howland, Tim Frazier, Ayman El-Menyar

**Affiliations:** 1Department of Surgery, Trauma and Vascular Surgery, Hamad General Hospital, Doha 3050, Qatar; halthani@hamad.qa; 2Clinical Research, Trauma and Vascular Surgery, Hamad General Hospital, Doha 3050, Qatar; mekkodathil@yahoo.co.uk; 3Department of Decision Sciences and Business Analytics, College of Business, Florida International University, Miami, FL 33199, USA; attila.hertelendy@georgetown.edu; 4Critical Care Paramedic, HMC Ambulance Service, Doha 3050, Qatar; ihowland@hamad.qa; 5Emergency and Disaster Management Program, Georgetown University, Washington, DC 20001, USA; tim.frazier@georgetown.edu; 6Department of Clinical Medicine, Weill Cornell Medical College, Doha 24144, Qatar

**Keywords:** rural, urban, trauma, emergency medical services, municipalities, Qatar

## Abstract

*Background*: Prehospital care provided by emergency medical services (EMS) plays an important role in improving patient outcomes. Globally, prehospital care varies across countries and even within the same country by the geographic location and access to medical services. We aimed to explore the prehospital trauma care and in-hospital outcomes within the urban and rural areas in the state of Qatar. *Methods*: A retrospective analysis was conducted utilizing data from the Qatar National Trauma Registry for trauma patients who were transported by EMS to a level 1 trauma center between 2017 and 2018. Data were analyzed and compared between urban and rural areas and among the different municipalities in which the incidents occurred. *Results:* Across the study duration, 1761 patients were transported by EMS. Of that, 59% were transported from an urban area and 41% from rural areas. There were significant differences in the on-scene time and total prehospital time as a function of urban and rural areas and municipalities; however, the response time across the study groups was comparable. There were no significant differences in blood transfusion, intubation, hospital length of stay, and mortality. *Conclusion*: Within different areas in Qatar, the EMS response time and in-hospital outcomes were comparable. This indicates that the provision of prehospital care across the country is similar. The prehospital and acute in-hospital care are accessible for everyone in the country at no cost. Understanding the differences in EMS utilization and prehospital times contributes to the policy development in terms of equitable distribution of healthcare resources.

## 1. Introduction

Prehospital care is provided by the emergency medical services (EMS) and comprises ground and helicopter EMS (GEMS and HEMS). EMS plays an important role in the provision of emergency care to critically ill or injured patients. The “golden hour” is a fundamental principle of prehospital trauma care and underlines the importance of resuscitation, stabilization, and rapid transport to the emergency care facility [1]. Evidence suggests that the provision of advanced emergency medical care during the brief prehospital phase reduces mortality and morbidity of trauma patients [2,3,4,5].

Prehospital care varies across countries and may not necessarily be the same within the same country as evident from the urban–rural differences reported across the globe. When compared to urban settings, the longer prehospital response time and on-scene time in rural settings were associated with increased mortality among trauma patients [3,6,7,8,9]. Increased prehospital mortality risk among patients from rural areas with low population density was demonstrated in a Norwegian study [8]. The authors further demonstrated that factors such as response time, time and distance of primary transport, limitation of specialized care, and rapid triage in rural areas contributed to the urban–rural differences [8].

In addition to timely prehospital and in-hospital care, the mechanism of injury contributes to the outcomes of trauma patients. Variations in mechanism of injury by urban and rural areas were also reported. Motor vehicle-related injuries were more common in rural municipalities, whereas self-harm and assault-related injuries were more frequent in metropolitan (urban) areas [8]. With regard to the prehospital phase, studies showed that there was no association between transport time and mortality in both rural and urban settings [9]. However, several studies showed that longer transport time was a predictor of longer hospital stay [10]. Additionally, rapid transport was found to be beneficial for traumatic brain injury patients, as well as patients with penetrating trauma [11].

As per the strategic recommendation by the Qatar Healthcare Facilities Master Plan 2013–2033, a general hospital should provide critical care services and a fully equipped ambulance service [12]. The Hamad Medical Corporation (HMC) is the country’s largest healthcare provider, with nearly 76% of inpatient healthcare services in Qatar in 2011 being provided by the HMC [13]. Therefore, reports from the HMC are essential for realizing the country’s healthcare aspirations. Trauma remains one of the main health system challenges in the country; 23% of deaths in the country are associated with trauma [13]. Provision of quality healthcare in both the hospital and the prehospital settings, based on evidence obtained through research, remains crucial. Although a few studies on prehospital care in Qatar have been published [14,15,16,17,18], information about EMS utilization based on geographical locations is lacking. The present study aims to explore the prehospital care of trauma patients and in-hospital outcomes within the urban and rural areas, as well as among the eight municipalities in the country. As Qatar is going to host the football World Cup in 2022, such a study will be of value for decision-makers to set evidence-based injury prevention strategies.

## 2. Materials and Methods

A retrospective analysis of the Qatar National Trauma Registry and ambulance service data for all trauma patients transported by EMS to the level 1 trauma center was conducted. The study duration was 1 year between 1 June 2017 and 31 May 2018. All trauma patients transported to hospital from the scene by EMS and who required hospitalization were included. Patients transferred from other hospitals by EMS and patients who were declared dead by EMS were excluded.

The trauma registry database participates in both the National Trauma Data Bank in the USA and the Trauma Quality Improvement Program of the Committee on Trauma by the American College of Surgeons. The registry data undergo internal and external validation on a regular basis. The HTC is the sole level 1 trauma center in Qatar which receives and treats all moderate to severely injured patients across the country free of charge. Each year, a Trauma Code (Level I, II, or III Trauma Criteria) is activated for nearly 2500 patients, and almost 1800 are admitted to the HTC. T1 (triage-1) represents patients who need immediate life-saving intervention, while T2 (triage-2) patients require intermediate or urgent intervention within 2–4 h and T3 (triage-3) patients need medical treatment that can be delayed safely [19].

Data extracted included mode of EMS transport (GEMS or HEMS), prehospital times (on-scene time and total prehospital time), time the incident occurred (weekdays or weekends; working hours or non-working hours), age, gender, nationality, underlying diseases, mechanism of injury (road traffic injuries, falls, and other), injured body regions (head, chest, spine, or abdomen), injury severity score (ISS), management (such as intubation and blood transfusion), and outcome (length of stays in intensive care unit (ICU) or hospital and in-hospital mortality). The collected data were compared between two groups, i.e., urban vs. rural.

There is no consensus for the definition of rural and urban areas worldwide, particularly in rapidly developing countries; however, urban areas usually have a higher quality/standard of living and greater population density than rural areas. Data on urban and rural areas in Qatar were obtained from the ambulance data in which the information about zones where the events occurred was captured. Zones are the second-highest level of the administrative divisions in Qatar following municipalities. There are eight municipalities and 98 zones, in which zones 1–69 considered urban areas and the remaining are rural areas (Figure 1) [20,21]. Among the eight municipalities (Ad Dawhah, Al Rayyan, Al Daayen, Umm Salal, Al Khor, Al Shamal, Al Shahaniya and Al Wakrah), Ad Dawhah and some parts of Al Rayyan are considered urban areas, while the others are considered rural areas. A comparative analysis of patient characteristics, management, and outcomes of trauma patients as a function of the eight municipalities was also performed as part of this study.


Zones 1–69 make up the area that can be defined as the Greater Doha area. This area consists of smaller but more densely populated zones, both business and residential, and represents the most rapidly expanding part of the country with extensive construction projects underway [22]. The remaining zones, which are defined as rural are larger but less populated zones. The travel distance from the northernmost point of the country, Zone 79, to Doha, the location of the only level 1 Trauma center, is 110 km, approximately 1 h and 20 min by car. The travel time from the southernmost point of the country, Zone 97, to Doha is 95 km or around 1 h and 10 min by car. The country has seen the development of highways to connect these rural areas to Doha, increasing their accessibility and popularity as places for recreational activities on both land and sea [23].

The state of Qatar is a rapidly developing country that has an area of 11,437 km^2^, with a population (≥15 years of age) of about 2.4 million in 2019 where 91% of the total workforce constitutes foreign nationals [24]. Almost 11% of the population in Qatar constitutes citizens. According to the World Bank data, the proportion of the rural population is just 0.8% of the total population, which is the 10th lowest in the world [25]. Kuwait (355 miles away from Qatar) is the only other country in the Middle East and North African region with a lower rural population, fifth lowest rural population in the world, having a larger size (17,818 km^2^) and bigger population (4.2 million). As mentioned earlier, the HTC is the only level 1 trauma center in Qatar, located in Doha, which is the capital city. Injuries are assessed by ambulance services, and care for patients with minor injuries is provided by the nearest emergency departments of hospitals located in different areas of Qatar. These hospitals include Hamad General Hospital (Ad Dawhah municipality), Al Khor hospital (Al-Khor municipality), the Cuban Hospital (Al-Shahaniya municipality), Hazm Mebareek General Hospital and Sidra Medicine (both at Al-Rayyan municipality), and Al-Wakra hospital (Al-Wakrah municipality). Patients who need tertiary-level care following assessment at the scene or emergency departments of the nearest hospitals are transferred to the HTC by ground or helicopter EMS. The Ruwais area in Al-Shamal municipality (northern tip of Qatar) is the farthest place, approximately 107 km from the HTC.

The Hamad Medical Corporation Ambulance Service (HMCAS) is the sole national ambulance service for Qatar and provides prehospital care to the entire population in the country. The HMCAS operates in coordination with the National Command Center (NCC) [26], where all 999 emergency calls are received and triaged. HMCAS is a two-tiered service comprising ambulance paramedics (AP) and critical care paramedics (CCP); it employs around 1300 clinical and support staff, and it has 200 ambulances, 22 rapid response vehicles, 16 bicycles, and a fleet of three helicopters. HMCAS responds to over 100,000 emergency calls a year [25,26,27]. Ambulances are staffed by two APs, and the critical care rapid response vehicles are staffed by a CCP and a critical care assistant (CCA). CCPs respond to the highest-priority cases, including major trauma, with a wide scope of practice, including prehospital rapid sequence induction (RSI), administering tranexamic acid (TXA) for major trauma, and provision of narcotic analgesia. The scope of practice is regularly reviewed to reflect international best practice guidelines.

Initial information from emergency calls is captured by the computer-assisted dispatch (CAD) system known as “Najem” which is a Unified Geospatial Infrastructure, a bilingual (Arabic/English) web-based application [27,28]. In addition to the incident details and location, this dispatching application provides an overview of the location of responding vehicles and their availability status. The HMCAS follows a “hub and spoke model” of deployment, in which the hub comprises supervisory officers and other administrative and support functions, whereas the spokes are small standby ambulance stations positioned in strategic areas across the country. Presently, the distribution model of HMCAS divides Qatar into six hubs, each with five to seven spoke stations.

The total prehospital time in this study was defined as the duration between the EMS notification time and the hospital arrival time. Response time was the difference between EMS notification time and the EMS arrival on scene time. On-scene time was the time duration between EMS arrival at the scene and the departure from the scene. The three most severely injured body regions according to Abbreviated Injury Scale (AIS) scores were squared and added together to estimate the Injury Severity Score (ISS), which provides an overall score for polytrauma [29]. The ISS ranges from 0 to 75 where 1–8 is minor, 9–15 is moderate, 16–24 is serious, 25–49 is severe, 50–74 is critical, and 75 is nonsurvivable [29].

Statistical analysis: Data were presented as proportions, means ± standard deviation (SD), or medians with interquartile range (IQR) where appropriate. The chi-square test was used to compare proportions between the categorical groups. Normality of continuous variables was checked using the Kolmogorov–Smirnov test. Continuous variables were compared using Student’s *t*-test for two groups or ANOVA for >2 groups, for parametric data. Mann–Whitney U test and Kruskal–Wallis test were used for nonparametric data, where applicable. A two-tailed *p*-value <0.05 was considered statistically significant. Data analysis was carried out using the Statistical Package for Social Sciences version 21 (SPSS Inc., Chicago, IL, USA).

This study followed the Consolidated Standards of Reporting Trials (CONSORT) (Figure 2).

## 3. Results

In the study duration, 1761 patients were transported to the HTC by EMS, of which 59% were transported from an urban area and 41% from rural areas. Table 1 shows the overall results and comparative analyses based on urban and rural areas.

Table 2 shows the results of a comparative analysis between non-Qatari and Qatari patients in the present study. The transportation time, severity of injury, and outcome were comparable.

Table 3 demonstrates patient characteristics, prehospital course, and in-hospital outcomes by municipalities in Qatar.

There are eight stadiums for the upcoming FIFA (Fédération Internationale de Football Association) 2022 World Cup in five municipalities (four stadiums in Doha and one each in Al Rayyan, Al Daayen, Al Wakrah, and Al Khor).

HEMS utilization for trauma patient transport was 8% of the total; it was significantly higher in rural areas (20%), and it was most frequently used in the Al-Shamal municipality (71%). Overall, the median response time, on-scene time, and total prehospital time were 6, 21, and 72 min, respectively. The response times across the study groups were comparable. The scene time and total prehospital times were significantly higher in rural areas, especially in the Al-Shamal municipality. The majority of events occurred on weekdays; however, the probability of events occurring on weekends was higher in rural areas, particularly in the Al-Wakrah municipality. Both males and females were more likely to get injured in urban areas compared to rural areas.

Road traffic-related injuries were most frequent, followed by falls, with the vast majority of victims being non-Qatari young males. The analysis of mechanism of injury in rural areas found that “other” injuries including sports and recreational injuries were more common, followed by road traffic-related trauma. This was more prevalent in the Al-Wakrah municipality. The Umm-Salal municipality was noted for its higher proportion of road traffic trauma. There were no significant differences in head, chest, or abdominal injuries; however, the number of spinal injuries reported in rural areas was significantly higher.

Although the pattern of mechanism of injuries and injured body regions differed significantly by urban–rural areas and by the municipalities, there were no significant differences in the ISS and trauma activation level between the study groups. In addition, there were no significant differences in management such as blood transfusion and intubation and in the outcomes such as hospital or ICU length of stay, as well as in-hospital mortality. Overall, in-hospital mortality was 8%.

## 4. Discussion

The present study is the first of its kind from the state of Qatar, which examined the characteristics of trauma patients, EMS utilization, and in-hospital outcomes as a function of the location of an incident, i.e., by urban and rural areas and by the municipalities in Qatar. This study highlights the importance of prehospital care and ambulance service resource distribution within Qatar, as well as guiding the focus for future injury prevention strategies, particularly as Qatar prepares to host the FIFA 2022 World Cup. The HTC prehospital services including transportation and acute in-hospital care are accessible for everyone in the country at no cost regardless of the location and nationality. The study was based on the national trauma registry and ambulance service data, which revealed that there were significant differences in the prehospital intervals such as on-scene times and total prehospital times between urban and rural areas, as well as municipalities; however, response times across all study groups were comparable. The comparable response time across the study groups suggests that the deployment and access to prehospital care is equitable. The differences in on-scene and total prehospital times reported in the present study may be associated with the distance to the only tertiary care facility in the country, as well as significant differences in the pattern of injuries, especially the differences in mechanism of injuries and injured body regions. As there were no significant differences in the provision of in-hospital emergency care, such as blood transfusion and intubation, there were consequently no differences in in-hospital outcomes in terms of duration of hospital stay and mortality.

Interestingly, the study provided important information regarding the differences in the pattern of injuries that occurred in urban vs. rural areas and by municipality, which will have a role in strengthening injury prevention efforts in Qatar. Notably, Qatari nationals were more likely to be involved in events occurring in rural areas. Injuries that occurred in Al-Wakra and Al-Shamal municipalities were characterized by higher proportions of sport- and recreational-related injuries, as well as spinal injuries. These patients were generally younger in age. In addition to the future potential injury prevention aspects of this study, information on EMS resource utilization by municipalities obtained from this study could contribute to future resource utilization strategies. HEMS utilization was more common in rural areas when compared to urban areas; the Al-Shamal municipality recorded a high utilization of HEMS. This is due to the fact that the HEMS operational area is limited to rural areas.

Variations in prehospital care access based on urban and rural settings are evident in the medical literature. Although there is a paucity of information about such variations in Qatar, a previous observational pilot study demonstrated variations in response times. On the basis of an analysis of 394 high-priority calls received by the ambulance service, Wilson et al. demonstrated that rural setting response times were usually longer compared to the urban setting [28]. The authors reported an overall median average response time of 5 min and 32 s, with an average urban response time of 5 min and 15 s and an average rural response time of 6 min and 22 s [28]. This study was a pilot observational study conducted over a period of 2 days and, therefore, might not have captured all possible issues. In our study, the median response time was 7 min in urban areas and 6 min in rural areas; this difference was not statistically significant.

In an earlier study looking at prehospital time intervals in Qatar, Al-Thani et al. reported that the overall total prehospital time was 70 min and the on-scene time was 24 min [17]. The authors compared the prehospital times, patient characteristics, and outcomes on the basis of trauma activations. T1 patients were more frequently injured in road traffic accidents and had head and chest injuries with a greater severity of injury (median ISS = 22), as well as longer on-scene time (27 min) and reduced total prehospital times (65 min). In addition, longer on-scene time was associated with higher mortality in T1 patients, whereas total prehospital time was not. The study concluded that survival of the injured patients depends on the prehospital and in-hospital settings and the injury characteristics [17].

The present study demonstrated that the scene time and total prehospital time were longer in EMS transports from rural areas. Notably, this study also demonstrated that HEMS utilization was more frequent in rural areas. This is most likely due to the severity of injuries and the difficulty in GEMS transportation due to geographical distance and access issues, particularly in recreational-related injuries in the desert. Al-Thani et al. further demonstrated that patients transported by HEMS were characterized by a greater proportion of traumatic brain injury and had a greater injury severity when compared to patients transported by GEMS [18]. These facts are in line with our findings regarding prolonged on-scene and total prehospital times in rural regions. Of note, our study revealed that there were no statistically significant differences in management or outcomes of these patients. Since the present study excluded hospital cases dead on arrival from the analysis, information regarding the variations in prehospital mortality by location is unavailable. This may have provided a more in-depth understanding regarding the severity of the incidents that occur in the rural areas.

The urban–rural differences in terms of response time, on-scene time, transport rates, transport time, and survival rate were previously discussed by several authors globally. Shorter response times in rural areas were reported by a number of studies [30,31,32], whereas others demonstrated that EMS responses to urban patients were quicker than those to rural patients [5,6,33,34,35]. Longer response times associated with rural EMS transport were primarily due to geographical distances, ambulance type and resource availability, EMS locations, physical condition of roads, and transport infrastructure [33,34,36,37].

Similar to the findings of the present study, two studies reported that rural on-scene time was significantly higher than that in urban settings [5,6]. Additionally, four studies showed significantly longer total prehospital times in rural than urban settings [5,6,32,36]. Unlike the response time, the reason behind the prolonged on-scene time in rural areas is not clear. Factors such as the severity of injury and stabilizing interventions performed may contribute to prolonged on-scene time, along with other factors such as the number of responders and time required to safely prepare the patient for transport. In addition, increased injury severity, advanced intravenous device requirement, mode of transport, and the complexity of injuries also contribute to the prolonged on-scene time [33,34,37]. Longer total prehospital transport times with incidents in rural regions were mainly associated with longer travel distances, availability of HEMS, road conditions, and traffic congestion. Moreover, access to specialized trauma care may not be readily available in rural areas.

Studies also reported increased prehospital mortality risk among rural patients [5,33,36,38]. However, our study demonstrated that prehospital mortality risk was comparable across the urban–rural settings. Fatovich et al. revealed a significantly higher risk of mortality in rural areas when compared to urban areas [36]. Nordberg et al. reported higher 30-day survival among urban patients than rural trauma patients [38]. Masterson et al. demonstrated that, when compared to rural patients, urban patients were more likely to be discharged alive from hospitals [33]. Grossman et al. reported that rural patients had significantly higher risk of mortality before arrival when the response time was over 30 min [5]. Differences in survival by urban–rural settings found in previous studies could be partly explained by the speed of endotracheal intubation and the prompt management of pulseless electrical activity, both of which are time-dependent with respect to outcome [39]. However, a previous study from Qatar on the impact of prehospital intubation revealed that prehospital intubation was associated with a higher mortality when compared to intubation in the emergency room [14]. Establishment of level II trauma centers outside Doha in addition to regular auditing of the transportation times and configuration of the hub and spoke model will further improve the outcome of trauma in rural areas.

Limitations: This study was retrospective, which is one of its limitations. Several cases had missing information from prehospital records; however, trauma registry data provide all relevant information on patient characteristics, injury characteristics, management, and outcomes during the course of hospitalization. Notably, this study is representative of the country as the data were retrieved from the nationwide trauma database in the state of Qatar. Occupational-related injury in different zones was not discussed as it was out of the scope of this study. Patients transported from rural areas by EMS to HTC may not necessarily represent the people residing in rural areas, and many of them may be travelers involved in leisure activities by visiting remote desert areas or coastal regions. It is evident from our study that injuries which occurred in Al-Wakra and Al-Shamal municipalities were mostly sports- and recreational-related injuries. This study could be useful for injury prevention strategies for the upcoming football world cup 2022 in Qatar.

## 5. Conclusions

Differences in access to prehospital care by location within a country cannot be attributed solely to geographical distances. The severity of injury sustained may result in increased total prehospital times and time to definitive care. Although there were differences in urban/rural prehospital times except for response times, the comparable in-hospital outcomes may indicate that the provision of prehospital care in both rural and urban settings is identical. Understanding the differences in EMS utilization and prehospital course may contribute to future policy development on equitable distribution of quality care.

## Figures and Tables

**Figure 1 ijerph-18-04016-f001:**
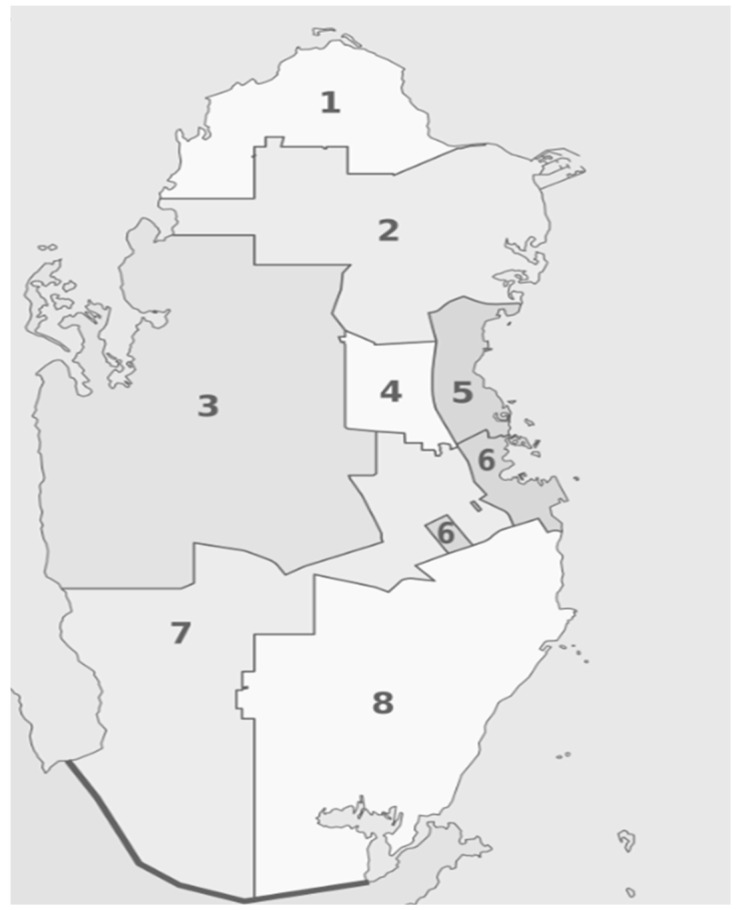
Map of the eight municipalities of Qatar in 2015.

**Figure 2 ijerph-18-04016-f002:**
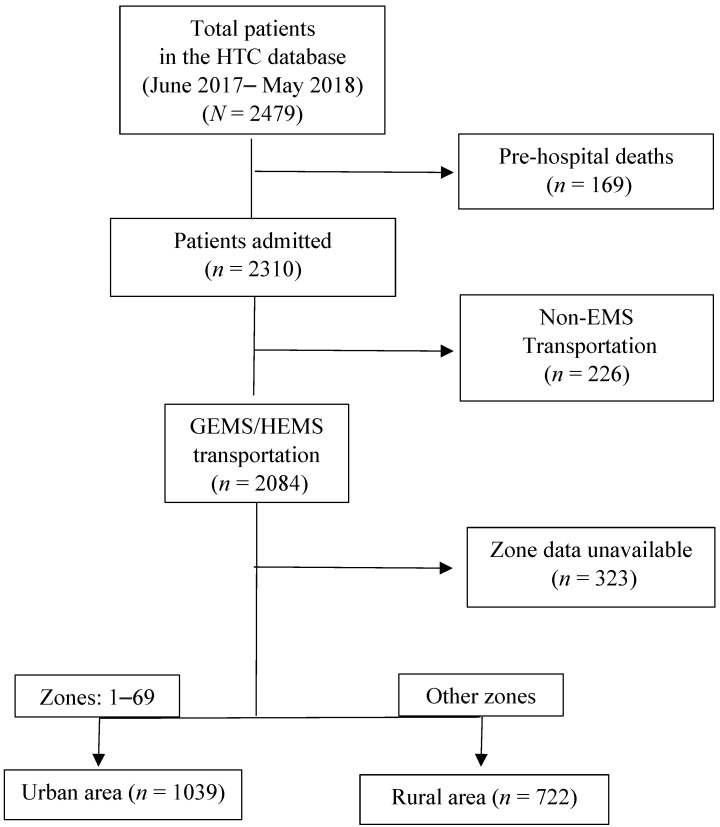
CONSORT (Consolidated Standards of Reporting Trials) flow diagram of emergency medical services (EMS)-transported patients to the Hamad Trauma Center (HMC), Qatar, by urban and rural areas. GEMS: ground EMS; HEMS: helicopter EMS.

**Table 1 ijerph-18-04016-t001:** Characteristics and outcomes of trauma patients transported by EMS from urban and rural locations.

	Overall (*n* = 1761)	Urban (*n* = 1039, 59.0%)	Rural (*n* = 722, 41.0%)	*p*
Transport mode				0.001
• GEMS	1616 (91.8)	1035 (99.6)	581 (80.5)	
• HEMS	145 (8.2)	4 (0.4)	141 (19.5)	
Median response time	6 (IQR 4–10)	7 (IQR 4–10)	6 (IQR 4–10)	0.25
Median scene time (IQR) min	21 (IQR 14–31)	19.5 (IQR 13–28)	23 (IQR 15–37)	0.001
Median prehospital time (IQR) min	72 (IQR 55–94)	64 (IQR 49–80)	87 (IQR 67–112)	0.001
Weekdays	1474 (83.8)	897 (86.5)	577 (79.9)	0.002
Weekends	285 (16.2)	140 (13.5)	145 (20.1)	
Age	30.9 ± 15.8	31.8 ± 15.9	31.0 ± 14.6	0.32
Gender				0.001
• Male	1530 (86.9)	880 (84.7)	650 (90.0)	
• Female	231 (13.1)	159 (15.3)	72 (10.0)	
Nationality				0.002
• Qatari	336 (19.1)	168 (16.2)	168 (23.3)	
• Others	1422 (80.9)	868 (83.8)	554 (76.7)	
Mechanism of injuries				0.001
• Road traffic injuries	916 (52.0)	529 (50.9)	387 (53.6)	
• Falls	473 (26.9)	315 (30.3)	158 (21.9)	
• Others	372 (21.1)	171 (18.8)	201 (24.5)	
Injured body regions				
• Head	494 (28.1)	295 (28.4)	199 (27.6)	0.70
• Chest	572 (32.5)	328 (31.6)	244 (33.8)	0.32
• Spine	375 (21.3)	185 (17.8)	190 (26.3)	0.001
• Abdomen	290 (16.5)	177 (17.0)	131 (15.7)	0.44
Median Injury Severity Score	10 (IQR 5–17)	10 (IQR 5–17)	10 (IQR 5–17)	0.45
Trauma activation				0.07 for all
• Level 1	343 (19.5)	200 (19.4)	143 (19.9)	
• Level 2	1190 (67.6)	690 (66.9)	500 (69.7)	
• Other	228 (12.9)	142 (13.7)	95 (10.4)	
Blood transfusion	290 (16.5)	166 (16.0)	124 (17.2)	0.51
Massive transfusion protocol	75 (4.3)	48 (4.6)	27 (3.7)	0.37
Intubation	369 (21.0)	225 (21.7)	144 (19.9)	0.39
Median length of stay (IQR) days				
• ICU	4 (IQR 2–10)	4 (IQR 2–11)	4 (IQR 2–9)	0.37
• Hospital	3 (IQR 1–9)	3 (IQR 1–9)	4 (IQR 1–11)	0.09
In-hospital mortality	142 (8.1)	78 (7.5)	64 (8.9)	0.30

HEMS: helicopter emergency medical services; GEMS: ground emergency medical services; IQR: interquartile range; ICU, intensive care unit.

**Table 2 ijerph-18-04016-t002:** Comparative analysis between non-Qatari and Qatari patients.

	Non-Qatari Patients	Qatari Patients	*p*-Value
Municipalities			0.001 for all
• Ad-Dawhah	439 (32.7)	72 (21.6)	
• Al Rayyan	433 (32.3)	108 (32.4)	
• Al Daayen	69 (5.1)	23 (6.9)	
• Umm Salal	44 (3.3)	9 (2.7)	
• Al-Khor	65 (4.8)	26 (7.8)	
• Al-Wakrah	199 (14.8)	52 (15.6)	
• Al-Shahaniya	77 (5.7)	30 (9.0)	
Mechanism of injury			0.001 for all
• Road traffic injuries	647 (48.2)	205 (61.6)	
• Falls	391 (29.1)	70 (21.0)	
• Others	304 (22.7)	198 (59.5)	
Median prehospital time (IQR) min	72 (56.0–93.5)	70 (51–94)	0.40
Median Injury Severity Score (IQR)	10 (5–17)	10 (5–17)	0.57
Median hospital length of stay (IQR) days	5 (2–12)	5 (2–15)	0.82
In-hospital mortality	55 (4.1)	19 (5.7)	0.20

**Table 3 ijerph-18-04016-t003:** Comparative analysis of prehospital and in-hospital data of trauma patients by municipalities in Qatar (*N* = 1750) *.

	Ad Dawhah (Doha) (*n* = 522; 29.8)	Al Rayyan (*n* = 568; 32.5)	Al Daayen (*n* = 97, 5.5)	Umm Salal (*n* = 58; 3.3)	Al Khor (*n* = 96; 5.5)	Al Shamal (*n* = 31; 1.8)	Al Shahaniya (*n* = 115; 6.6)	Al Wakrah (*n* = 263; 15)	*p*-Value
Transport mode									0.001
➢ GEMS	520 (99.6)	558 (98.2)	96 (99.0)	56 (96.6)	79 (82.3)	9 (29.0)	85 (73.9)	202 (76.8)	
➢ HEMS	2 (0.4)	10 (1.8)	1 (1.0)	2 (3.4)	17 (17.7)	22 (71.0)	30 (26.1)	61 (23.2)	
Median response time (IQR) min	7 (IQR 4–11)	6 (IQR 4–10)	7 (IQR 5–10)	5 (IQR 3–8)	6 (IQR 4-10)	6 (IQR 4–7.3)	8 (IQR 4–14)	6 (IQR 3–10)	0.13
Median scene-time (IQR) min	19 (IQR 13.0–28.0)	20.0 (IQR 13.0–30.0)	23.0 (IQR 15.0–35.0)	23.0 (IQR 14.0–32.0)	22.0 (IQR 15.3–32.3)	39 (IQR 26.0–54.0)	23.5 (IQR 15.0–42.0)	23.0 (IQR 14.0–37.0)	0.001
Median prehospital time (IQR) min	60.0 (IQR 46.0–79.8)	67.0 (IQR 53.0-82.0)	75.0 (IQR 64.0–90.0)	80.0 (IQR 62.0–80.0)	92.0 (IQR 81.0–115.0)	114.0 (IQR 102.0–137.5)	89.0 (IQR 72.0–110.0)	90.0 (IQR 67.0–115.0)	0.001
Weekends	70 (13.4)	77 (13.6)	14 (14.4)	11 (19.0)	14 (14.6)	6 (19.4)	27 (23.5)	65 (24.7)	0.001
Weekdays	451 (86.6)	490 (86.4)	83 (85.6)	47 (81.0)	82 (85.4)	25 (80.6)	88 (76.5)	198 (75.3)	
Age	33.4 ± 15.6	30.1±15.9	32.2 ± 13.1	32.1 ± 11.6	32.5 ± 15.9	26.8 ± 14.5	30.0 ± 12.7	31.0 ± 16.0	0.02
Gender									0.076
➢ Male	442 (84.7)	488 (85.9)	85 (87.6)	50 (86.2)	87 (90.6)	29 (93.5)	110 (95.7)	228 (86.7)	
➢ Female	80 (15.3)	80 (14.1)	12 (12.4)	8 (13.8)	9 (9.4)	2 (6.5)	5 (4.3)	35 (13.3)	
Mechanism of injury									0.001 for all
➢ RTI	250 (47.9) 155	298 (52.5)	55 (56.7)	44 (75.9)	55 (57.3)	18 (58.1)	72 (62.6)	115 (43.7)	
➢ Falls	(29.7) 117	177 (31.2)	23 (23.7)	9 (15.5)	23 (24.0) 18	5 (16.1)	21 (18.3)	58 (22.1)	
➢ Others	(22.4)	93 (16.4)	19 (19.6)	5 (8.6)	(18.8)	8 (25.8)	22 (19.1)	90 (34.2)	
Median Injury Severity Score (IQR)	10 (IQR 5–19)	10 (IQR 5–17)	12 (IQR 5–19)	14 (IQR 5–19)	12 (IQR 5–17)	10 (IQR 5–19.5)	10 (IQR 5–21)	10 (IQR 5–17)	0.095
Trauma activation									0.381
Level 1	107 (20.5)	106 (18.9)	21 (21.9)	15 (25.9)	15 (15.6)	6 (19.4)	26 (23.0)	45 (17.2)	
Level 2	342 (65.6)	378 (67.4)	66 (68.8)	35 (60.3)	75 (78.1)	23 (74.2)	75 (66.4)	190 (72.5)	
Other	72 (13.8)	77 (13.7)	9 (9.4)	8 (13.8)	6 (6.3)	2 (6.5)	12 (10.6)	27 (10.3)	
Median length of stay (IQR) days									
ICU	4 (IQR 2–11)	3.5 (IQR 2–10)	5 (IQR 2–15)	6 (IQR 3–8.75)	3 (IQR 2–4.25)	5 (IQR 2–13.25)	5 (IQR 2–12.25)	3 (IQR 2–7.5)	0.539
Hospital	3 (IQR 1–10)	3 (IQR 1–9)	5 (IQR 2–14.75)	5 (IQR 1–16.5)	3.5 (IQR 2–10)	3.5 (IQR 1–15)	4 (IQR 1–10)	3 (IQR 1–10)	0.111
In-hospital mortality	35 (6.7)	46 (8.1)	12 (12.4)	7 (12.1)	11 (11.5)	2 (6.5)	11 (9.6)	17 (6.5)	0.346

* 11 cases had missing municipality information, HEMS: helicopter emergency medical services; GEMS: ground emergency medical services; IQR: interquartile range; ICU: intensive care unit; RTI: road traffic injury.

## Data Availability

Not applicable.

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
