# Peer review of "Emergency Medical Services (EMS) Transportation of Trauma Patients by Geographic Locations and In-Hospital Outcomes: Experience from Qatar"

_ijerph, 2021, doi:10.3390/ijerph18084016_

Round 1

Reviewer 1 Report

The authors do not present new findings in their study that will be useful to the scientific and professional public. 

The results of the study present already known truths, for example: „The study concluded that the survival of the injured patients depends on prehospital, and in-hospital settings and the injury characteristics.“ or  „The present study demonstrated that the scene-time and total prehospital time was longer in EMS transport from rural areas.“

The source data is not current, the data was taken from an older period 01/06/2017 and 31/05/2018. It would be appropriate to implement data from the current period 01/01/2020 and 31/12/2020.

Author Response

Thanks for your comment. Actually our region in the Arab Middle East is lacking such study. Therefore our study is unique in describing the EMS services based on the geographic location and outcomes. Such study is important to explore the prehospital infrastructure and to set in reality the required for injury prevention measures especially that Qatar is hosting a big event next year which is the football world cup . we agree that the source of data is not current but also it is not so outdated, we got the IRB approval in 2019 and then then the pandemic was the main concern. However, the data for 2018-2020 still not approved yet ,once approved we ,ofcourse , will work to publish it before the end of this year

Reviewer 2 Report

The paper is relatively short and seems to be rather a research note than a full research paper. Of course this is not negative remark, but as a reviewer I’m oblige to underline this comment. However the paper is very interesting and well prepared.

Because the Authors analyzed data and compared them between urban and rural area groups and among the different municipalities where the events occurred within the country, the Author(s) should firstly define separately the urban and rural areas (in general and in Qatar case) and add some spatial differences background.

I think, at least a map is missing. The map would be important for readers to know about localization of the municipalities involved into research (all localities listed in the Table 2 should be presented on the map).

The Author have to add some information and conclusions about accessibility which is one of the main factors of healthcare system from the spatial perspective. (these explanations would be helpful in understanding differences between urban and rural areas and be helpful for conclusions).

Some very minor technical mistake appeared in the text: e.g. line 113 - should be  (not km2).

Author Response

Thanks for your comments

We have added (There is no consensus for the definition of rural and urban area worldwide and particularly in the rapid developing countries; however, the urban usually has higher quality/standards of living and population density than that of the rural area. Data on urban and rural areas in Qatar were obtained from the ambulance data in which the information about zones where the events occurred is captured. Zones are the second-highest level of the administrative divisions of Qatar following municipalities. There are 8 municipalities and 98 zones in which zones 1 to 69 are in urban areas and the remaining are in rural areas (Fig. 1) [20,21]. Out of the eight municipalities (Ad Dawhah, Al Rayyan, Al Daayen, Umm Salal, Al Khor, Al Shamal, Al Shahaniya and Al Wakrah), Ad Dawhah and some parts of Al Rayyan are urban areas while the others are considered rural areas. A comparative analysis for patient characteristics, management, and outcomes of trauma patients by the 8 municipalities was also performed as part of this study).

Also map for the 8 municipalities added

The HTC is the sole level 1 trauma center in Qatar which receives and treats all moderate to severely injured patients across the country free of charge.

Correction for typo and errors were highlighted.

Reviewer 3 Report

Review: Emergency Medical Services (EMS) Transportation of Trauma Patients by Geographic Locations and In-Hospital Outcomes:  Experience from Qatar

General

The authors present an overview of the Qatari prehospital rescue system. No really new findings are presented. Furthermore, the authors do not follow the CONSORT reporting, this has to be changed.

Major concern

  1. Abstract: The conclusion is daring with indication of the provision of quality of prehospital care across the country. In fact, you show similar patient times and clinical outcomes, but this does not directly indicate the quality of care. This would require comparisons with other systems. At most one can speak of the same treatment in urban and rural areas.
  2. Method: According to CONSORT, a Flow Chart is missing. This would be particularly useful because in the discussion you speak of the exclusion of the deceased.
  3. Results: Both Table 1 and Table 2 are ambiguous with regard to the p values. What do these relate to in detail? For example first line: There is only one p value. Which values are compared there with one another?
  4. Results: Table 2: In the lower area, the table has been completely shifted and is not meaningful to read, please adapt.
  5. Table 2 line 185: The table caption is also incomprehensible. It has to be corrected.
  6. The limitation section is sparse. It is precisely here that there is talk of missing data, but not in terms of inclusion or exclusion in the method section, see Major concern 2
  7. Conclusion: see number 1.

Minor concern

  1. English needs improvement
  2. Line 39: Prehospital care IS provided…
  3. Throughout the text are often too many spaces in the entire text (e.g. Line 103, 105, 130, 157,…) Please check punctuation marks and commas
  4. Line 121 is incomprehensible: a fleet of three helicopters that responds to over 100,000 emergency calls... Probably all deployments of all life-saving appliances are meant, please clarify
  5. Definition of ISS: 16-24 serious, 50-74 critical. What about 25-49?

Author Response

General

The authors present an overview of the Qatari prehospital rescue system. No really new findings are presented. Furthermore, the authors do not follow the CONSORT reporting, this has to be changed.

Reply: Thanks. This is the first study in our region in the Arab middle east that addressed the prehospital care and outcomes in the urban and rural areas. This is actually needed to explore the need for further improvement especially in our country that will host the upcoming FIFA football world cup next year

We added CONSORT figure

Major concern

  1. Abstract: The conclusion is daring with indication of the provision of quality of prehospital care across the country. In fact, you show similar patient times and clinical outcomes, but this does not directly indicate the quality of care. This would require comparisons with other systems. At most one can speak of the same treatment in urban and rural areas. Reply: We do agree and we addressed that the care and its quality and outcome were comparable. All people are receiving the emergency care at our trauma center free of charge
  2. Method: According to CONSORT, a Flow Chart is missing. This would be particularly useful because in the discussion you speak of the exclusion of the deceased. Reply: CONSORT added
  3. Results: Both Table 1 and Table 2 are ambiguous with regard to the p values. What do these relate to in detail? For example first line: There is only one p value. Which values are compared there with one another? Reply: we revised it. P value is for all variable within the same raw
  4. Results: Table 2: In the lower area, the table has been completely shifted and is not meaningful to read, please adapt. Reply: thanks, revised
  5. Table 2 line 185: The table caption is also incomprehensible. It has to be corrected. Reply: revised
  6. The limitation section is sparse. It is precisely here that there is talk of missing data, but not in terms of inclusion or exclusion in the method section, see Major concern 2. Reply: Revised thanks
  7. Conclusion: see number 1. Reply: revised and correction was highlighted in color

Round 2

Reviewer 3 Report

Thank you for revising the manuscript. Overall, the presentation is significantly better, but some grammar and phrasing errors still remain. Furthermore, the graphic of Figure 2 should be revised, the print quality is clearly poor.

Major concern

  • Conclusion: The inserted sentence “The HTC is the only level 1 trauma center in Qatar which receives and treats all moderate to severely injured patients across the country free of charge.” in the conclusions should be removed again as this is a clarification, but not a conclusion from the study

Author Response

Thank you for revising the manuscript. Overall, the presentation is significantly better, but some grammar and phrasing errors still remain. Furthermore, the graphic of Figure 2 should be revised, the print quality is clearly poor.

Reply: figures improved

Major concern

  • Conclusion: The inserted sentence “The HTC is the only level 1 trauma center in Qatar which receives and treats all moderate to severely injured patients across the country free of charge.” in the conclusions should be removed again as this is a clarification, but not a conclusion from the study

Reply: the sentence has been removed from the conclusion